# Time to Death and Nursing Home Admission in Older Adults with Hip Fracture: A Retrospective Cohort Study

**DOI:** 10.3390/jcm14238603

**Published:** 2025-12-04

**Authors:** Yoichi Ito, Norio Yamamoto, Yosuke Tomita, Kotaro Adachi, Masaaki Konishi, Kunihiko Miyazawa

**Affiliations:** 1Pharmacy Department, Minato Medical Coop-Kyoritsu General Hospital, Nagoya 456-8611, Aichi, Japan; 2Department of Orthopedic Surgery, Minato Medical Coop-Kyoritsu General Hospital, Nagoya 456-8611, Aichi, Japan; 3Scientific Research WorkS Peer Support Group (SRWS-PSG), Osaka 541-0043, Osaka, Japan; 4Department of Physical Therapy, Faculty of Health Care, Takasaki University of Health and Welfare, Takasaki 370-0033, Gunma, Japan; 5Postgraduate Training Department, Minato Medical Coop-Kyoritsu General Hospital, Nagoya 456-8611, Aichi, Japan; 6Department of Orthopaedic Surgery, Nagano Red Cross Hospital, Nagano 380-0928, Nagano, Japan

**Keywords:** hip fracture, femoral neck fracture, trochanteric fracture, death, nursing home

## Abstract

**Background**: Hip fractures in older adults are sentinel events linked to high mortality and functional decline. Few studies have quantified long-term survival probabilities, standardized mortality ratios (SMRs), and risks of new nursing home admission alongside patient-related predictors. **Methods**: We retrospectively analyzed 355 patients aged ≥ 60 years who underwent hip fracture surgery at a general hospital in Japan (2020–2024). Primary outcomes were mortality and new nursing home admission. Survival probabilities and remaining life expectancy were estimated, and SMRs were calculated using age- and sex-matched national data. Cox regression identified independent predictors. **Results**: Mean age was 84 years; 76% were female. Mortality probabilities at 1, 2, and 3 years were 23%, 41%, and 60%, respectively; SMRs consistently exceeded 9. Median remaining life expectancy was 260 days. New nursing home admissions occurred in 42%, with cumulative probabilities of 16%, 27%, and 35% at 1, 2, and 3 years, respectively, showing a rapid rise within 9 months. Independent predictors of mortality were delayed surgery, higher Charlson Comorbidity Index, and low Geriatric Nutritional Risk Index. Older age and failure to regain ambulatory ability at 3 months predicted institutionalization. **Conclusions**: Older adults with hip fractures face persistently high mortality and institutionalization risks, comparable to advanced malignancies or neurodegenerative diseases. Surgical timing, comorbidities, nutrition, and functional recovery critically influence prognosis and should guide perioperative care and discharge planning.

## 1. Introduction

Hip fractures are a major health concern among older adults, often leading to substantial functional decline and increased mortality [1]. Following a hip fracture, many patients experience reduced ambulatory ability and a loss of independence in activities of daily living (ADLs), which may result in long-term disability, institutionalization, or death [2]. Despite advances in surgical techniques and perioperative care, the overall prognosis remains poor, particularly among frail elderly individuals.

However, few studies have examined the temporal trajectories of key outcomes—such as postoperative mortality, new nursing home admission, and ADL decline—after discharge from acute care hospitals following hip fracture surgery [3,4]. Consequently, the longitudinal course of these outcomes remains poorly understood. Moreover, many previous studies have not accounted for critical patient-related factors, including comorbidities, nutritional status, and socioeconomic conditions, when analyzing postoperative outcomes in older adults.

Therefore, we conducted a retrospective cohort study to characterize the long-term prognosis of older adults undergoing hip fracture surgery, focusing on the temporal patterns of mortality and new nursing home admissions. Based on prior evidence of frailty-related vulnerability, we hypothesized that patient-related factors—including comorbidity burden, nutritional status, functional impairment, and socioeconomic conditions—would be independently associated with poorer long-term outcomes. Our objectives were to quantify the median survival times and absolute annual risks for these outcomes and to evaluate how these individual characteristics contribute to variations in prognosis.

## 2. Materials and Methods

### 2.1. Study Design and Setting

This single-center retrospective cohort study was conducted in accordance with the principles of the Declaration of Helsinki (1964 and its later amendments). The study adhered to the Strengthening the Reporting of Observational Studies in Epidemiology (STROBE) guidelines [5] (Appendix A). The study protocol was approved by the Institutional Ethics Committee of Kyoritsu General Hospital (approval no. 2024.11) and was preregistered on the Open Science Framework (https://osf.io/njzxm, accessed on 9 June 2025).

### 2.2. Participant Selection

Data were collected from consecutive patients who underwent surgery for proximal femoral fractures (hip fractures, AO/OTA types 31A and 31B) [6] at a general hospital in Japan between February 2020 and December 2024. Patients aged over 60 years were included. Exclusion criteria were as follows: multiple fractures involving the lower limbs, pathological fractures due to causes other than osteoporosis, and open fractures. When patients sustained bilateral fractures during the study period, only data from the first hip fracture were analyzed.

### 2.3. Variables

Orthopedic surgeons and pharmacists reviewed medical records and extracted the following variables: age, sex, height, weight, ambulatory ability, place of residence before the hip fracture (own home [living alone or not living alone], nursing home, or hospital), public assistance status, history of hip fracture, dementia, Charlson Comorbidity Index (CCI) [7], and serum albumin level at admission.

Ambulatory ability was categorized into four levels: no aid, use of a cane, use of a walker, and use of a wheelchair.

Living alone and receiving public assistance under the Japanese welfare system were included as indicators of socioeconomic status (SES).

As an indicator of nutritional status, the Geriatric Nutritional Risk Index (GNRI) was calculated using body weight, height, and serum albumin level at admission [8].

Surgical variables included the time from injury to surgery (surgical waiting time) and the type of surgical procedure performed.

### 2.4. Treatment Strategy

Surgical treatment was generally performed to relieve pain and restore ambulatory function, except when contraindicated for medical reasons or when patients opted for conservative treatment. Surgery under spinal anesthesia was planned as early as possible after admission when patients’ medical conditions allowed. In patients receiving antiplatelet or anticoagulant therapy, surgery was either delayed or performed under general anesthesia.

For femoral neck fractures, either internal fixation or hemiarthroplasty was selected based on fracture type and surgeon preference. Trochanteric fractures were treated with internal fixation using either a sliding hip screw or a cephalomedullary nail.

Postoperative physiotherapy—including bed mobilization and gait training—was initiated on the first postoperative day, and full weight-bearing was permitted immediately after surgery. Most patients were transferred to a rehabilitation hospital approximately one month postoperatively, after which they were discharged home or to a care facility depending on ambulatory ability and living environment.

### 2.5. Outcomes

All patients were included in the analysis regardless of the follow-up duration, as in-hospital mortality was part of the outcomes.

The primary outcomes were all-cause mortality and new admission to a nursing home during follow-up. The date of death was obtained from hospital records, including both in-hospital deaths and out-of-hospital deaths that were formally reported to the hospital. The number of days from surgery to death was recorded as the remaining life expectancy at surgery. Patients already residing in a nursing home at the time of their initial hip fracture were excluded from the nursing home analysis.

The secondary outcomes were contralateral hip fracture and postoperative ambulatory ability. Patients with a prior contralateral hip fracture at baseline were excluded. Ambulatory ability was assessed at 3 and 12 months postoperatively. Independent gait was defined as walking without personal assistance (no aid, cane, or walker). The proportion of patients who regained their pre-fracture ambulatory ability was calculated, excluding those who had used a wheelchair before injury.

### 2.6. Statistical Analysis

Categorical variables were expressed as counts and percentages, and continuous variables as medians with interquartile ranges (IQRs).

Probabilities of death and nursing home admission were calculated monthly after surgery. Mortality risk was evaluated using standardized mortality ratios (SMRs), calculated as the ratio of observed to expected deaths, adjusted for age, sex, and calendar year of surgery, based on official Japanese life table data [9]. An SMR > 1.0 indicated higher mortality than expected in the general population, and SMR < 1.0 indicated lower mortality. SMRs and 95% confidence intervals (CIs) were estimated for each postoperative month. We restricted the analysis to patients who were newly institutionalized during follow-up, and defined early institutionalization as admission within 3 months postoperatively. Mortality was compared between early vs. later institutionalization at 12, 24, and 36 months using Fisher’s exact test.

Missing data were handled using multiple imputation under the missing-at-random assumption. Twenty imputed datasets were generated using multiple imputation by chained equations, incorporating all covariates and outcomes [10].

Univariate logistic regression analyses (Model 1) were followed by multivariable analyses: Model 2 (adjusted for age and sex) and Model 3 (including all pre-specified covariates) [11].

Time-to-event outcomes were evaluated using Cox proportional hazards regression to estimate hazard ratios (HRs) and 95% CIs, with time zero defined as the date of surgery. Follow-up for mortality continued until death or the end of follow-up, and for new nursing home admission until the first admission or censoring at 3 years (1095 days) or the last known contact. Multivariable Cox models included the same covariates as Model 3.

Prespecified subgroup analyses were conducted by sex (female vs. male), ambulatory ability (independent [no aid, cane, walker] vs. dependent [wheelchair]), dementia (yes/no), GNRI-based classification (high risk: GNRI < 92 vs. low risk: GNRI ≥ 92) [8], socioeconomic status (living alone vs. not living alone) [3], and fracture laterality (unilateral vs. bilateral). Interaction terms were used to assess effect modification across subgroups.

Sensitivity analyses were performed to confirm robustness. For mortality, we excluded (i) deaths within 30 days after surgery, (ii) patients with dementia, and (iii) patients using a wheelchair preoperatively. For new nursing home admission, we excluded (i) admissions within 30 days after surgery, (ii) patients receiving public assistance, and (iii) performed complete-case analyses without imputation. All sensitivity analyses used the same covariate structure as Model 3.

All tests were two-tailed, and *p* < 0.05 was considered statistically significant. Analyses were conducted using R software (version 4.5.0; R Foundation for Statistical Computing, Vienna, Austria).

## 3. Results

After excluding six cases with multiple fractures involving the lower limbs, the final analysis included 355 patients (mean age, 85.2 years; standard deviation, 7.8), of whom 289 (81.4%) were women. Independent gait was observed in 86.1% of patients, with 48.7% walking without any aid (Table 1). Before surgery, 35.2% of patients lived in nursing homes. The median time from injury to surgery was 5.0 days (IQR, 2.0–11.0). Surgical treatments included internal fixation in 26 patients and bipolar hemiarthroplasty (BHA) in 155 patients for femoral neck fractures, and cannulated hip screws (CHS) in 106 patients, intramedullary nailing in 67 patients, and BHA in one patient for trochanteric fractures.

The mortality rate during the follow-up period was 21.7%, with a 1-year mortality rate of 23% (Table 1). The median remaining life expectancy after surgery was 260 days (IQR, 87–559). New nursing home admissions occurred in 42% of patients, with a median postoperative time of 149 days (IQR, 98–227). The incidence of contralateral hip fractures was 4.5%, with a median onset of 181 days after surgery (IQR, 78–338). At 3 months postoperatively, 25% of patients regained their pre-fracture ambulatory ability and 58% achieved independent gait. At 12 months, 30% had regained their pre-fracture ambulatory ability and 65% had achieved independent gait.

The probabilities of death and new nursing home admission increased progressively over time (Appendix A). The probability of death was 0.23 (95% CI, 0.17–0.28) at 1 year, 0.41 (95% CI, 0.33–0.49) at 2 years, and 0.60 (95% CI, 0.51–0.69) at 3 years after surgery. The probability of nursing home admission was 0.16 (95% CI, 0.11–0.21) at 1 year, 0.27 (95% CI, 0.20–0.35) at 2 years, and 0.35 (95% CI, 0.26–0.43) at 3 years. The probability of death increased at a nearly constant rate, whereas the probability of nursing home admission rose more rapidly during the first 9 months after surgery. At 12 months, early institutionalization showed a tendency toward higher mortality (OR 7.48, 95% CI 0.65–117.9, *p* = 0.062). Similar, though statistically non-significant, trends were observed at 24 and 36 months.

The standardized mortality ratio (SMR) for mortality risk exceeded 10 at 6 and 9 months after surgery but remained in the range of 9 at other time points (Appendix A). The highest SMR was observed at 9 months postoperatively, with a value of 10.2 (95% CI, 6.43–13.9).

Univariate analyses identified pre-fracture ambulatory ability, CCI, GNRI, and days from injury to surgery as significant predictors of mortality (Table 2, Appendix A). Univariate analyses of new nursing home admissions identified age and dementia as significant predictors (Table 3, Appendix A).

Multivariable logistic regression analysis (Model 3) identified male sex, delayed surgical timing, higher CCI scores, and high-risk GNRI as independent predictors of mortality (Appendix A). For new nursing home admissions, multivariable logistic regression analysis (Model 3) demonstrated that failure to regain ambulatory ability at 3 months postoperatively was a significant predictor (Appendix A).

Cox proportional hazards regression analysis identified delayed surgical timing as an independent predictor of mortality (Table 4; Figure 1). For new nursing home admissions, Cox regression (Model 3) demonstrated that failure to regain ambulatory ability at 3 months postoperatively was a significant predictor (Table 5; Figure 2).

Subgroup analyses using the Cox regression model for mortality showed that older age was associated with reduced survival among male patients, those with dementia, and those with bilateral fractures. Early surgery was associated with improved survival in the GNRI no-/low-risk subgroup and in patients with trochanteric fractures (Appendix A). For new nursing home admission, subgroup analyses of the adjusted Cox model revealed no significant effect modification across subgroups (Appendix A).

Sensitivity analyses for mortality and new nursing home admission confirmed the robustness of the primary findings (Appendix A).

## 4. Discussion

Our findings indicate that patients undergoing hip fracture surgery face substantial long-term risks of mortality and new nursing home admission. Mortality probabilities at 1, 2, and 3 years were 23%, 41%, and 60%, respectively, showing a steady increase over time. Mortality risk remained consistently elevated (SMR > 9) compared with the general population throughout the follow-up period. The incidence of new nursing home admission was 42%, with the greatest increase occurring within the first nine months after surgery. Delayed surgical timing, higher comorbidity burden, and poor nutritional status were associated with increased mortality, whereas advanced age and failure to regain ambulatory ability were significant predictors of new nursing home admission.

Mortality after hip fracture surgery remained persistently high in our cohort, with survival probabilities comparable to those observed in advanced malignancies or neurodegenerative diseases such as amyotrophic lateral sclerosis [12,13]. A recent global review reported a pooled 1-year mortality of 22.0%, with regional variation—mean rates of 23.3% in Europe and 17.9% in Asia [14]. Our 1-year mortality of 23% therefore lies at the upper end of the expected range. With respect to excess mortality, previous studies have shown standardized mortality ratios (SMRs) of 8.0 in a large Spanish cohort and approximately 7 in a Norwegian cohort [15,16], which are broadly similar to the SMRs exceeding 9 observed in our population. These geographic differences suggest that system-level factors may partly influence post-fracture mortality patterns across countries.

The risk of new nursing home admission after hip fracture surgery was substantial, increasing rapidly during the early postoperative months and continuing to rise over time. One year after surgery, only 40% of patients maintained independent living at home. This admission rate is comparable to that observed in chronic progressive diseases such as dementia and stroke, making it remarkably high for an acute traumatic injury [17,18]. These findings suggest that hip fracture often triggers a trajectory of functional decline rather than representing an isolated orthopedic event. This highlights its profound and lasting impact on the functional independence of older adults and its role as a turning point toward institutionalization.

Independent predictors of mortality were delayed surgery, greater comorbidity burden, and poor nutritional status. Subgroup analyses revealed particularly poor survival among older men, patients with dementia, and those with bilateral fractures, indicating heightened vulnerability in these groups. Early surgery was associated with improved outcomes, particularly in patients without nutritional impairment and those with trochanteric fractures. Although the survival benefits of early surgery have been reported previously, several studies have suggested that its effects may be attenuated in frail or nutritionally compromised patients [19]. This potential interaction underscores the importance of individualized surgical decision-making, where timely intervention may be most beneficial for patients with preserved physiological reserves. Overall, our findings emphasize the need for optimizing perioperative care through prompt surgery, comprehensive management of comorbidities, and proactive nutritional support.

Independent predictors of new nursing home admission were older age and failure to regain ambulatory ability at 3 months. Functional decline is a key determinant of institutionalization. For patients living with family, dementia or mobility loss may exceed the family’s caregiving capacity, whereas in those living alone, impaired ambulation may preclude continued independent living. These findings highlight the importance of early, intensive rehabilitation and targeted social support interventions aimed at preserving mobility and preventing institutionalization.

This study has several strengths. First, to our knowledge, this is the first study to evaluate median survival times and absolute annual risks of mortality and new nursing home admission after hip fracture surgery. We provided detailed monthly trajectories of these outcomes, together with changes in SMRs, which are directly applicable to risk communication and informed consent in clinical practice. We also examined other clinically relevant outcomes, such as ambulatory ability and contralateral hip fracture. Second, we incorporated important patient-related factors, including socioeconomic status (living alone, public assistance) and nutritional status (GNRI), thus offering a comprehensive understanding of prognostic determinants. Third, the study adhered to the STROBE guidelines and a preregistered protocol, ensuring transparency and methodological rigor.

This study also has limitations. First, the median time from injury to surgery was 5.0 days in our cohort, which is longer than the 2-day standard recommended by international guidelines [20]. This delay may be attributable to mild symptoms from occult fractures leading to delayed presentation, limited medical access, and the need to postpone surgery for patients with comorbidities or those receiving anticoagulant therapy until spinal anesthesia could be safely administered in the absence of a full-time anesthesiologist. Second, remaining life expectancy may have been underestimated because it was calculated only for patients whose date of death was confirmed in hospital records, including both in-hospital deaths and out-of-hospital deaths that were formally reported back to the hospital. Conversely, the incidence of mortality may have been underestimated because deaths occurring outside the hospital could not be identified. This limitation also affects the interpretation of the high SMRs and short median remaining life expectancy observed in this study. Because patients with unverified out-of-hospital deaths contributed follow-up time but not death events, the survival time was truncated and SMRs may have been inflated. Therefore, these estimates should be interpreted as conservative lower-bound survival estimates rather than the true prognosis for all patients after hip fracture. Third, the generalizability of our findings is limited because the study was conducted at a single general hospital in Japan, and caution is required when extrapolating the results to other institutions or populations.

As a research implication, future studies should evaluate tailored interventions to determine whether early surgery or socioeconomic support programs, when combined with individual risk factors such as age, comorbidities, nutritional status, and functional status, can mitigate the risks of mortality and nursing home admission identified in this study. Moreover, larger multicenter prospective studies are warranted to confirm the generalizability of our findings and to strengthen causal inferences.

## 5. Conclusions

Older adults with hip fractures face persistently high risks of mortality and new nursing home admission after surgery. Surgical timing, nutritional status, and functional recovery are key determinants of prognosis. Incorporating socioeconomic, nutritional, and functional factors into perioperative management and discharge planning may help improve long-term outcomes and support patient-centered care.

## Figures and Tables

**Figure 1 jcm-14-08603-f001:**
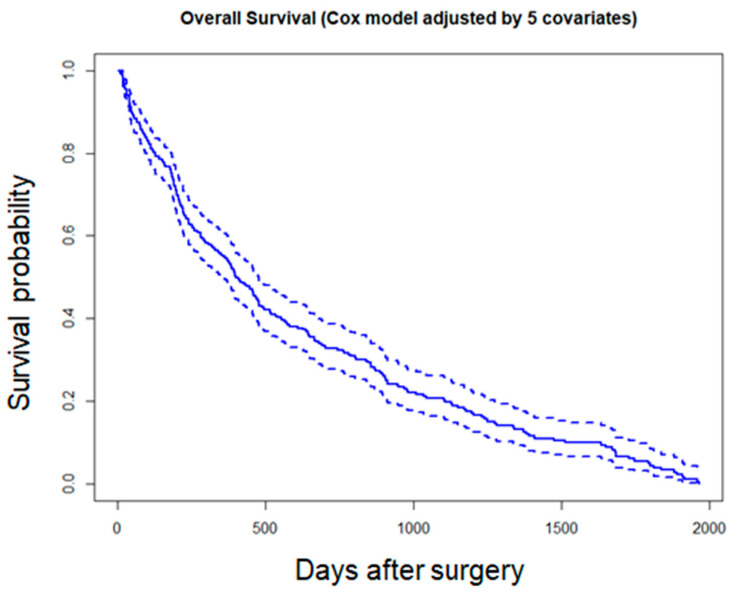
Cox regression survival curve (Model 3): mortality. Solid lines indicate adjusted probability, and dotted lines indicate 95% confidence intervals.

**Figure 2 jcm-14-08603-f002:**
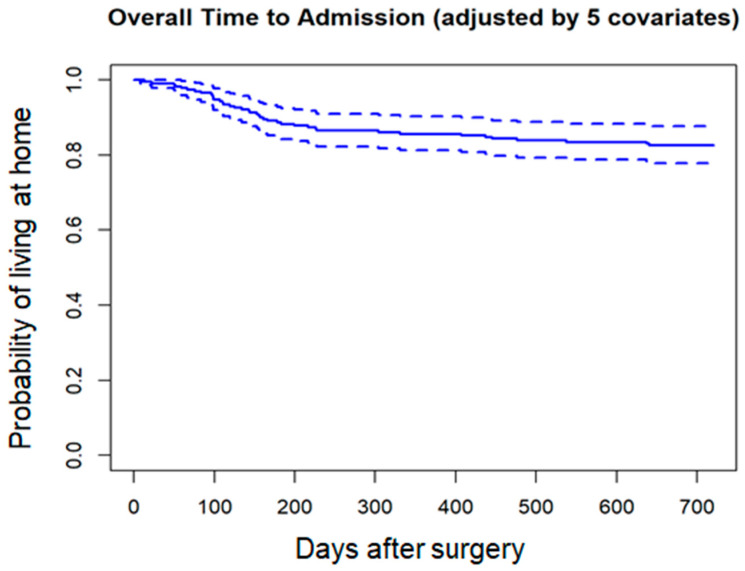
Cox regression survival curve (Model 3): new nursing home admission. Solid lines indicate adjusted probability, and dotted lines indicate 95% confidence intervals.

**Table 1 jcm-14-08603-t001:** Patient characteristics and outcomes.

		Total (*n* = 355)	Femoral Neck Fracture (*n* = 180)	Trochanteric Fractures (*n* = 175)
Age, years		87 (81, 90)	84 (80, 90)	88 (84, 92)
Women, *n* (%)		289 (81.4)	146 (81.8)	143 (81.7)
Height, cm		148 (142, 155)	149.0 (144, 156)	147.0 (140, 153)
Weight, kg		43.5 (38.0, 50.9)	42.8 (38.0, 50.9)	43.8 (38.5, 51.4)
BMI, kg/m^2^		19.9 (17.6, 22.8)	19.3 (20.0, 22.0)	20.6 (17.9, 23.7)
Pre-fracture ambulatory ability, *n* (%)	No aid	173 (48.7)	100 (55.6)	73 (41.7)
	Cane	63 (17.7)	23 (12.8)	40 (22.9)
	Walker	70 (19.7)	28 (15.6)	42 (24.0)
	Wheelchair	49 (13.8)	29 (16.1)	20 (11.4)
Pre-fracture residence, *n* (%)	Own home (living alone)	59 (16.6)	25 (13.9)	34 (19.4)
	Own home (non-living alone)	156 (43.9)	84 (46.7)	72 (41.1)
	Nursing home	125 (35.2)	63 (35.0)	62 (35.4)
	Hospital	15 (4.2)	8 (4.4)	7 (4.0)
Public assistance recipient, *n* (%)		30 (8.5)	18 (10.0)	12 (6.9)
History of hip fracture, *n* (%)		29 (8.2)	15 (8.3)	14 (8.0)
Dementia, *n* (%)		136 (38.3)	68 (37.8)	68 (38.9)
Charlson comorbidity index		1.0 (1.0, 2.0)	1.0 (1.0, 2.0)	1.0 (1.0, 3.0)
Albumin, g/dL		3.7 (3.3, 3.9)	3.7 (3.3, 4.0)	3.6 (3.2, 3.9)
Geriatric Nutritional Risk Index, *n* (%)	Major risk (GNRI < 82)	74 (20.8)	44 (24.4)	30 (17.1)
	Moderate risk (GNRI 82 to <92)	100 (28.2)	47 (26.1)	53 (30.3)
	Low risk (GNRI 92 to ≤98)	70 (19.7)	34 (18.9)	36 (20.6)
	No risk (GNRI > 98)	111 (31.3)	55 (30.6)	56 (32.0)
Days to surgery from injury		5.0 (2.0, 11.0)	7 (3.0, 12.0)	4 (2.0, 7.0)
Surgery type, *n*			BHA 154, CCHS 23, CHS 3	CHS 100, Nailing 75
Mortality, *n* (%)		77 (21.7)	39 (21.7)	38 (21.7)
Remaining life expectancy, days		260 (87, 559)	288.0 (97.5, 657.5)	224.5 (75.0, 498.5)
New nursing home admission, *n* (%)		42 (11.8)	16 (8.9)	26 (14.9)
New nursing home admission, days		148.5 (98.3, 226.8)	158.0 (99.3, 251.0)	143.5 (98.3, 226.8)
Incidence of contralateral hip fracture, *n* (%)		16 (4.5)	8 (4.4)	8 (4.6)
Contralateral hip fracture, days		180.5 (77.5, 337.8)	146.0 (77.5, 257.5)	225.5 (80.8, 337.8)
Ambulatory ability at 3 months postoperatively, *n* (%)	No aid	40 (11.3)	33 (18.3)	7 (4.0)
	Cane	55 (15.5)	25 (13.9)	30 (17.1)
	Walker	56 (15.8)	26 (14.4)	30 (17.1)
	Wheelchair	141 (39.7)	62 (34.4)	79 (45.1)
	Unknown	63 (17.7)	34 (18.9)	29 (16.6)
Ambulatory ability regained at 3 months postoperatively, *n* (%)		65 (25.3)	43 (34.1)	22 (16.8)
Independent gait regained at 3 months postoperatively, *n* (%)		148 (57.6)	81 (64.3)	67 (51.1)
Ambulatory ability at 12 months postoperatively, *n* (%)	No aid	31 (8.7)	24 (13.3)	7 (4.0)
	Cane	44 (12.4)	25 (13.9)	19 (10.9)
	Walker	29 (8.2)	10 (5.6)	19 (10.9)
	Wheelchair	69 (19.4)	32 (17.8)	37 (21.1)
	Unknown	182 (51.3)	89 (49.4)	93 (53.1)
Ambulatory ability regained at 12 months postoperatively, *n* (%)		48 (30.4)	30 (37.0)	18 (23.4)
Independent gait regained at 12 months postoperatively, *n* (%)		103 (65.2)	58 (71.6)	45 (58.4)
Follow-up days		290.0 (109, 693.5)	330 (111.0, 843.5)	268 (104.5, 616.5)

Data are presented as number (%) or median (interquartile range). Geriatric Nutritional Risk Index, GNRI; Bipolar Hemiarthroplasty, BHA; Cannulated Cancellous Hip Screw, CCHS; Compression Hip Screw, CHS.

**Table 2 jcm-14-08603-t002:** Univariate analysis of risk factors associated with mortality.

		Total (*n* = 355)	Survivors (*n* = 278)	Death (*n* = 77)	*p*-Value
Age, years		87 (81, 90)	86.5 (60, 101)	87.0 (68, 100)	0.17
Women, *n* (%)		289 (81.4)	232 (83.5)	57 (74.0)	0.08
BMI, kg/m^2^		19.9 (17.6, 22.8)	20.1 (17.8, 23.2)	19.2 (15.8, 21.7)	0.70
Charlson comorbidity index		1.0 (1.0, 2.0)	1.0 (0.0, 2.0)	2.0 (1.0, 2.0)	<0.01
Albumin, g/dL		3.7 (3.3, 3.9)	3.7 (3.4, 4.0)	3.4 (3.1, 3.8)	3.02
Geriatric Nutritional Risk Index, *n* (%)	No risk (GNRI > 98)	111 (31.3)	94 (33.8)	17 (22.1)	<0.01
Surgery from injury, days		5.0 (2.0, 11.0)	4.0 (2.0, 9.8)	8.0 (4.0, 13.0)	<0.01

Data are presented as number (%) or median (interquartile range). Geriatric Nutritional Risk Index, GNRI.

**Table 3 jcm-14-08603-t003:** Univariate analysis of risk factors associated with new nursing home admission.

		Total (*n* = 230)	Not New Admitted (*n* = 189)	New Admitted (*n* = 41)	*p*-Value
Age, years		86 (80, 90)	84 (60, 98)	90 (69, 99)	<0.01
Women, *n* (%)		181 (78.7)	145 (76.7)	36 (87.8)	0.14
Pre-fracture residence, *n* (%)	Own home (non-living alone)	59 (25.6)	131 (69.3)	25 (61.0)	0.74
Dementia, *n* (%)		113 (49.1)	39 (20.6)	16 (39.0)	0.02

Data are presented as number (%) or median (interquartile range).

**Table 4 jcm-14-08603-t004:** Cox regression models for mortality.

		Hazard Ratio (95% CI)	*p*-Value
Model 2	Age	0.96 (0.93–1.00)	0.08
	Sex, man	0.48 (0.25–0.91)	0.02
Model 3	Age	1.01 (0.99, 1.03)	0.53
	Sex, man	0.96 (0.92–1.00)	0.06
	Early surgery	0.48 (0.25–0.95)	0.03
	CCI	2.09 (1.09–4.27)	0.03
	GNRI: no/low risk	0.77 (0.65–0.91)	<0.01

Charlson comorbidity index, CCI; Geriatric Nutritional Risk Index, GNRI.

**Table 5 jcm-14-08603-t005:** Cox regression models for new nursing home admissions.

		Hazard Ratio (95% CI)	*p*-Value
Model 2	Age	0.91 (0.87–0.97)	<0.01
	Sex, man	1.58 (0.56–4.44)	0.38
Model 3	Age	0.94 (0.89–1.00)	0.07
	Sex, man	1.46 (0.48–4.42)	0.50
	Dementia	0.53 (0.22–1.24)	0.14
	Living with someone before fracture	1.52 (0.65–3.56)	0.33
	Ambulatory ability regained at postoperative 3 months	2.88 (1.28–6.46)	0.01

## Data Availability

Due to institutional and ethical restrictions, the individual-level patient data are not publicly available. De-identified aggregated data supporting the main results are available from the corresponding author upon reasonable request.

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
