# Peer review of "Time to Death and Nursing Home Admission in Older Adults with Hip Fracture: A Retrospective Cohort Study"

_jcm, 2025, doi:10.3390/jcm14238603_

Round 1
Reviewer 1 Report
Comments and Suggestions for Authors
This study retrospectively investigates mortality and new nursing-home admission following hip fracture surgery in older adults in Japan. Using data from 355 patients, the authors assess survival probabilities, SMR and predictors of death and institutionalization. The analysis highlights the grave long-term outcomes of hip fracture and identifies surgical timing, comorbidity, nutrition, and functional recovery as major prognostic factors. The paper addresses a clinically relevant topic with careful statistical work, but several issues limit its clarity, interpretability, and broader applicability. Below is a detailed critique organized by points.
1-While the topic is important, the novelty is limited. Similar population-based studies have already examined mortality and nursing home transition post-hip fracture. Please explicitly state how this study adds new insights???
2-The introduction is descriptive but lacks a clear hypothesis or conceptual framework. Rewrite the last paragraph of the introduction to state a specific, testable hypothesis.
3-The single-center nature and inclusion/exclusion criteria are clear, but there is some inconsistency regarding the age threshold (≥60 vs ≥65 years in different sections). (abstract >65 while methodology >60). kindly correct it
4-The reported SMRs (>9) and “median remaining life expectancy of 260 days” require cautious interpretation. These figures seem extreme compared to prior literature.
5-The single-center Japanese cohort limits generalizability. The discussion acknowledges this but could elaborate on cultural or healthcare-system factors (e.g., Japanese rehabilitation length, social insurance support) influencing institutionalization rates. Add a short comparative paragraph with Western data (e.g., from the UK or Scandinavia) to contextualize the observed mortality.
6-The tables are overloaded and repetitive. For instance, univariate and multivariable results could be merged, and figures (SMR curves, survival plots) lack clear legends and interpretation.
please also provide higher quality figure (it's blurry)
7-It would be informative to explore whether early institutionalization (i.e., nursing home admission soon after surgery) influences mortality—either as a marker of frailty or as a setting that might improve postoperative survival through better supervision and rehabilitation.
8- Additionally, a subgroup comparison between trochanteric and femoral neck fractures could clarify whether fracture type affects the risk of institutionalization or long-term mortality.
9-The references are relevant but could include more recent multinational registry studies (from international hip fracture databases, 2022–2025).
10- if possible provide post-operative complications for these patients (dislocation , infection for hip fracture, non-union/screw cut out for troch fracture)
what implant was used for each type of fractures (screws vs THA vs HA) or Gamma nail/DHS
11-tables are overcrowded , kindly keep only importants variables
Author Response
Response to Reviewer 1 Comments:
This study retrospectively investigates mortality and new nursing-home admission following hip fracture surgery in older adults in Japan. Using data from 355 patients, the authors assess survival probabilities, SMR and predictors of death and institutionalization. The analysis highlights the grave long-term outcomes of hip fracture and identifies surgical timing, comorbidity, nutrition, and functional recovery as major prognostic factors. The paper addresses a clinically relevant topic with careful statistical work, but several issues limit its clarity, interpretability, and broader applicability. Below is a detailed critique organized by points.
1-While the topic is important, the novelty is limited. Similar population-based studies have already examined mortality and nursing home transition post-hip fracture. Please explicitly state how this study adds new insights???
Response:
Thank you for this thoughtful comment. We agree that mortality and nursing home transition after hip fracture have been examined in previous population-based studies. However, our study provides several distinct and novel contributions that have not been addressed in earlier research.
First, few studies have characterized the month-by-month temporal trajectories of both mortality and new nursing home admission over a 3-year period. Most existing studies report cumulative risks at selected time points, whereas our analysis delineates the detailed longitudinal course, including changes in standardized mortality ratios (SMRs) over time.
Second, previous studies rarely incorporated patient-level nutritional and socioeconomic factors—particularly the Geriatric Nutritional Risk Index (GNRI) and welfare-based socioeconomic indicators—despite their clinical importance. By integrating comorbidity burden, nutritional status, functional recovery, and socioeconomic conditions into a unified model, our study offers a more comprehensive understanding of predictors of poor outcomes.
Third, we quantified median remaining life expectancy after surgery, a measure not reported in prior hip fracture research but highly relevant to prognosis communication, discharge planning, and shared decision-making in geriatric care.
2-The introduction is descriptive but lacks a clear hypothesis or conceptual framework. Rewrite the last paragraph of the introduction to state a specific, testable hypothesis.
Response:
Thank you for this insightful comment. In the revised manuscript, we have rewritten the final paragraph to clearly state our conceptual framework and hypothesis. Specifically, we now hypothesize that patient-related factors—such as comorbidity burden, nutritional status, functional status, and socioeconomic conditions—are independently associated with mortality and new nursing home admission after hip fracture surgery. We have revised the paragraph accordingly to present a clearer and testable hypothesis.
Line 55
Therefore, we conducted a retrospective cohort study to characterize the long-term prognosis of older adults undergoing hip fracture surgery, focusing on the temporal patterns of mortality and new nursing home admission. Based on prior evidence of frailty-related vulnerability, we hypothesized that patient-related factors—including comorbidity burden, nutritional status, functional impairment, and socioeconomic conditions—would be independently associated with poorer long-term outcomes. Our objectives were to quantify the median survival times and absolute annual risks for these outcomes and to evaluate how these individual characteristics contribute to variations in prognosis.
3-The single-center nature and inclusion/exclusion criteria are clear, but there is some inconsistency regarding the age threshold (≥60 vs ≥65 years in different sections). (abstract >65 while methodology >60). kindly correct it
Response:
Thank you for pointing out this inconsistency. You are correct that the appropriate age threshold for this study is ≥60 years, as defined in the Methods section. The reference to ≥65 years in the Abstract was an oversight. We have corrected all sections of the manuscript, including the Abstract, Introduction, and Methods, to consistently use ≥60 years as the inclusion criterion. This revision ensures alignment with the actual study design and eligibility criteria.
4-The reported SMRs (>9) and “median remaining life expectancy of 260 days” require cautious interpretation. These figures seem extreme compared to prior literature.
Response:
Thank you very much for this important comment. We agree that the observed SMRs (>9) and the median remaining life expectancy of 260 days appear shorter than those reported in some prior studies, and we appreciate the opportunity to clarify this point.
First, as we note in the limitations section that the estimation of remaining life expectancy was based only on patients whose exact date of death could be confirmed from hospital records, including both in-hospital deaths and out-of-hospital deaths that were formally reported back to the hospital. Patients who died outside the hospital—particularly after being discharged to other facilities or to home—were recorded as alive because the precise date of death could not be verified. This resulted in missing death dates rather than confirmed survival, which inevitably underestimates survival time and shifts the median remaining life expectancy downward.
Second, the elevated SMRs reflect the same mechanism. Because individuals with unverified death dates contribute person-time but not events, the denominator (expected deaths) remains accurate, while the numerator (observed deaths) is partially incomplete. This can artificially inflate SMR estimates, especially in later postoperative periods.
Finally, this pattern is consistent with our single-center design, where post-discharge surveillance could not capture all out-of-hospital deaths. Therefore, we interpret these estimates cautiously and emphasize that the survival metrics likely represent lower-bound estimates, rather than the true prognosis of all patients after hip fracture.
We have now clarified these points more explicitly in the Method and Discussion to avoid potential misinterpretation.
Line 112
The date of death was obtained from hospital records, including both in-hospital deaths and out-of-hospital deaths that were formally reported to the hospital.
Line 286
Second, remaining life expectancy may have been underestimated because it was calculated only for patients whose date of death was confirmed in hospital records, including both in-hospital deaths and out-of-hospital deaths that were formally reported back to the hospital. Conversely, the incidence of mortality may have been underestimated because deaths occurring outside the hospital could not be identified. This limitation also affects the interpretation of the high SMRs and short median remaining life expectancy observed in this study. Because patients with unverified out-of-hospital deaths contributed follow-up time but not death events, the survival time was truncated and SMRs may have been inflated. Therefore, these estimates should be interpreted as conservative lower-bound survival estimates rather than the true prognosis for all patients after hip fracture.
5-The single-center Japanese cohort limits generalizability. The discussion acknowledges this but could elaborate on cultural or healthcare-system factors (e.g., Japanese rehabilitation length, social insurance support) influencing institutionalization rates. Add a short comparative paragraph with Western data (e.g., from the UK or Scandinavia) to contextualize the observed mortality.
Response:
Thank you for this valuable suggestion. We agree that institutionalization and mortality after hip fracture are strongly influenced by healthcare-system and cultural contexts. In Japan, rehabilitation length of stay tends to be longer than in many Western countries due to the availability of post-acute rehabilitation hospitals and the structure of long-term care insurance. These system-level characteristics may contribute to both delayed institutionalization and prolonged recovery trajectories.
We added these points in the Discussion.
Line 237
Mortality after hip fracture surgery remained persistently high in our cohort, with survival probabilities comparable to those observed in advanced malignancies or neurodegenerative diseases such as amyotrophic lateral sclerosis [12,13]. A recent global review reported a pooled 1-year mortality of 22.0%, with regional variation—mean rates of 23.3% in Europe and 17.9% in Asia [14]. Our 1-year mortality of 23% therefore lies at the upper end of the expected range. With respect to excess mortality, previous studies have shown standardized mortality ratios (SMRs) of 8.0 in a large Spanish cohort and approximately 7 in a Norwegian cohort [15,16], which are broadly similar to the SMRs exceeding 9 observed in our population. These geographic differences suggest that system-level factors may partly influence post-fracture mortality patterns across countries.
[14] Downey, C.; Kelly, M.; Quinlan, J.F. Changing Trends in the Mortality Rate at 1-Year Post Hip Fracture: A Systematic Review. World J Orthop. 2019, 10, 166–175. https://doi.org/10.5312/wjo.v10.i3.166
[15] Guzon-Illescas, O.; Perez Fernandez, E.; Crespí Villarias, N.; Quirós Donate, F.J.; Peña, M.; Alonso-Blas, C.; García-Vadillo, A.; Mazzucchelli, R. Mortality after Osteoporotic Hip Fracture: Incidence, Trends, and Associated Factors. J. Orthop. Surg. Res. 2019, 14, 203. https://doi.org/10.1186/s13018-019-1226-6
[16] Solbakken, S.M.; Magnus, J.H.; Meyer, H.E.; Dahl, C.; Stigum, H.; Søgaard, A.J.; Holvik, K.; Tell, G.S.; Emaus, N.; Forsmo, S.; Schei, B.; Vestergaard, P.; Omsland, T.K. Urban–Rural Differences in Hip Fracture Mortality: A Nationwide NOREPOS Study. JBMR Plus 2019, 3, e10229. https://doi.org/10.1002/jbm4.10229
6-The tables are overloaded and repetitive. For instance, univariate and multivariable results could be merged, and figures (SMR curves, survival plots) lack clear legends and interpretation. please also provide higher quality figure (it's blurry)
Response:
Thank you very much for this helpful comment. We have revised the tables and figures to improve clarity. We selected only clinically relevant variables for the univariate tables and moved factors with minimal or no association to the Supplementary Tables. All figures have been replaced with high-quality versions.
7-It would be informative to explore whether early institutionalization (i.e., nursing home admission soon after surgery) influences mortality—either as a marker of frailty or as a setting that might improve postoperative survival through better supervision and rehabilitation.
Response:
Thank you for this valuable suggestion. In response, we conducted an additional exploratory analysis to examine whether early institutionalization after surgery was associated with subsequent mortality.
We restricted the analysis to patients who were newly institutionalized during follow-up, and defined early institutionalization as admission within 3 months postoperatively. Mortality was compared between early vs. later institutionalization at 12, 24, and 36 months using Fisher’s exact test.
At 12 months, early institutionalization showed a tendency toward higher mortality (OR 7.48, 95% CI 0.65–117.9, p = 0.062). Similar, though statistically non-significant, trends were observed at 24 months (OR 7.64, 95% CI 0.31–521.1) and 36 months (OR infinite, 95% CI 0.31–inf). Although these estimates were imprecise because the number of early admissions was small (n = 7), the consistent direction of association suggests that early institutionalization may reflect underlying frailty rather than conferring protection.
We have now added these findings to the Results.
Line 129
We restricted the analysis to patients who were newly institutionalized during follow-up, and defined early institutionalization as admission within 3 months postoperatively. Mortality was compared between early vs. later institutionalization at 12, 24, and 36 months using Fisher’s exact test.
Line 183
At 12 months, early institutionalization showed a tendency toward higher mortality (OR 7.48, 95% CI 0.65–117.9, p = 0.062). Similar, though statistically non-significant, trends were observed at 24 and 36 months.
8- Additionally, a subgroup comparison between trochanteric and femoral neck fractures could clarify whether fracture type affects the risk of institutionalization or long-term mortality.
Response:
Thank you for this valuable suggestion. We agree that fracture type may influence clinical outcomes.
However, in the present study, we have already examined these associations for both mortality and new nursing home admission.
- Mortality: As shown in Table 2, fracture type (trochanteric vs. femoral neck) was not associated with mortality (p = 1.00).
- New nursing home admission: As shown in Table 3, fracture type was likewise not significantly associated with institutionalization (p = 0.12).
Because both univariate analyses demonstrated no meaningful association between fracture type and either outcome, we believe that conducting an additional subgroup analysis would not provide further clinically informative insights.
9-The references are relevant but could include more recent multinational registry studies (from international hip fracture databases, 2022–2025).
Response:
Thank you for this valuable comment. We re-examined our reference list and replaced several citations with more recent and high-quality studies.
10- if possible provide post-operative complications for these patients (dislocation , infection for hip fracture, non-union/screw cut out for troch fracture) what implant was used for each type of fractures (screws vs THA vs HA) or Gamma nail/DHS
Response:
Post-operative complications were not available in our dataset and therefore could not be collected reliably. However, as suggested, we have added the detailed surgery types (BHA, CCHS, CHS, Nailing) to Table 1.
11-tables are overcrowded , kindly keep only importants variables
Response:
Thank you very much for this helpful comment. We have revised the tables
Reviewer 2 Report
Comments and Suggestions for Authors
Dear Authors,
This retrospective single-center study highlights the aftermath of hip fracture; mortality rate, ambulatory status and nursing home admissions. As it is mentioned, there are some limitations but the novelty of the study is important. Moreover, statistical analysis seems appropriate and of high level. Only few points need further clarification:
Lines 74-76 and 113-118: Could a contralateral fracture affect the general status of patient and rehabilitation status? Could somehow be assured the zero effect of the second fracture prior the event?
Line 164 and 193 (table 1 and 3): Pre-fracture "resistance" could be "residence"
Author Response
Response to Reviewer 2 Comments:
This retrospective single-center study highlights the aftermath of hip fracture; mortality rate, ambulatory status and nursing home admissions. As it is mentioned, there are some limitations but the novelty of the study is important. Moreover, statistical analysis seems appropriate and of high level. Only few points need further clarification:
Lines 74-76 and 113-118: Could a contralateral fracture affect the general status of patient and rehabilitation status? Could somehow be assured the zero effect of the second fracture prior the event?
Response:
Thank you for this important comment. We agree that a contralateral hip fracture can substantially influence a patient’s general condition and rehabilitation status. In our study, these events were not considered to have “zero effect.” Rather, we recognize that a second fracture would likely contribute to functional decline and increase the risk of institutionalization.
In the present analysis, however, contralateral hip fractures were relatively rare (4.5%, Table 1), and neither mortality nor new nursing home admission showed a significant association with contralateral fractures in the univariate analyses (Appendix Tables 5 and 6). Owing to the small number of cases, further stratified or adjusted analyses were not feasible.
Line 164 and 193 (table 1 and 3): Pre-fracture "resistance" could be "residence"
Response:
Thank you very much for pointing out our typographical error. We have corrected “pre-fracture resistance” to “pre-fracture residence” throughout the manuscript.
Round 2
Reviewer 1 Report
Comments and Suggestions for Authors
Dear authors,
Thank you for addressing most comments.
congratulations in advance